# UVMap-ID: A Controllable and Personalized UV Map Generative Model

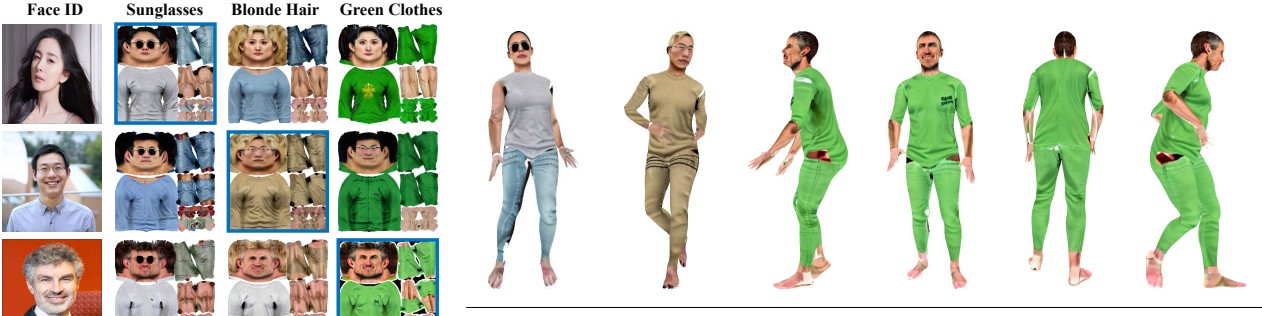

**Face ID**  **Sunglasses**  **Blonde Hair**  **Green Clothes**

**Rendered Results on SMPL**

**Figure 1: Our method can synthesize high-quality textures while enabling a controllable and personalized generation with the given text prompts and Face ID (Left). The textures can be directly applied to SMPL meshes [29] (Right).**

## ABSTRACT

Recently, diffusion models have made significant strides in synthesizing realistic 2D human images based on provided text prompts. Building upon this, researchers have extended 2D text-to-image diffusion models into the 3D domain for generating human textures (UV Maps). However, some important problems about UV Map Generative models are still not solved, i.e., how to generate personalized texture maps for any given face image, and how to define and evaluate the quality of these generated texture maps. To solve the above problems, we introduce a novel method, UVMap-ID, which is a controllable and personalized UV Map generative model. Unlike traditional large-scale training methods in 2D, we propose to fine-tune a pre-trained text-to-image diffusion model which is integrated with a face fusion module for achieving ID-driven customized generation. To support the finetuning strategy, we introduce a small-scale attribute-balanced training dataset, including high-quality textures with labeled text and Face ID. Additionally, we introduce some metrics to evaluate the multiple aspects of the textures. Finally, both quantitative and qualitative analyses demonstrate the effectiveness of our method in controllable and personalized UV Map generation.

## CCS CONCEPTS

• **Computing methodologies** → **Rasterization**; **Neural networks**; *Motion capture*; **Artificial intelligence**; **Rasterization**; **Texturing**; *Reconstruction*.

## KEYWORDS

Generative Model, Diffusion Model, 3D Avatar Generation, Multi-Modal Generation

## 1 INTRODUCTION

The development of 3D human models has garnered significant attention in recent years, owing to its versatile applications across various domains, including filmmaking, video games, augmented reality/virtual reality (AR/VR), and human-robot interaction. Among the myriad tasks essential for crafting digital humans, texture synthesis stands out as a pivotal element in achieving the photorealistic quality of 3D avatars. However, creating 3D textures in the traditional computer graphics pipeline is time-consuming and labor-intensive. Thus, it is important to utilize generation techniques to design diverse texture maps automatically.

Texture (UV map) generation has been a focus in previous approaches for tasks such as 3D face and human reconstruction. These methods leverage generators from Generative Adversarial Networks (GANs) to estimate textures either in an unsupervised [9, 45, 52, 59] or supervised [24, 25] manner. Subsequently, the texture estimation model is integrated into the avatar fitting stage. Nonetheless, these methods are limited in generating novel textures and need more support for controllable generation.

Large-scale text-to-image diffusion models [36, 38], nowadays, have been proven very effective over cross-model generation tasks, which should mainly attributed to the scalable 2D image-text data pairs along with large-scale parallel computation. Yet we notice that the lack of large-scale 3D texture data makes training high-quality texture generative models quite challenging. Inspired by the pre-trained strategy of DreamBooth, SMPLitex [5] has employed a few texture maps (UV defined by SMPL [29]) to fine-tune a pretrained text-to-image diffusion model. It has been observed that this approach enables the synthesis of texture maps while supporting its foundation text-driven task. However, the inability of SMPLitex to

support personalized texture generation poses a significant limitation on their approach, particularly in applications where user customization is crucial. Personalized texture generation enables the tailoring of textures to specific individual preferences, fostering a comprehensive experience in 3D applications, including avatars, VR, and gaming. Besides personalization, evaluating the quality of generated textures within the UV space remains an unresolved challenge, leaving more space for research.

In this paper, we introduce the UVMap-ID method, a UV map generation model that supports ID-driven personalized generation tasks. Specifically, we fine-tune a pretrained text-to-image diffusion model using a small-scale training dataset. In contrast to 2D personalized methods [7, 46, 49, 56] that necessitate large-scale training data in 2D methods, our dataset, which is attribute-balanced (i.e., "Race and Gender"), comprises around 750 image-ID pairs: the textures map with annotated text prompts, the corresponding portrait faces. To enable the ability of ID-driven personalized generation, we extend the stable diffusion with an additional face fusion module. Moreover, we introduce some corresponding metrics to evaluate the quality of generated textures from multiple aspects, i.e., fidelity, structure preservation, ID preservation, and text-image alignment. Remarkably, our model achieves high-quality and diverse texture synthesis within just several hours of training, while also supporting controllable and personalized synthesis with the user-provided image ID.

In summary, our contributions are as follows:

- We are the first to propose a controllable and personalized UV map generative model capable of synthesizing diverse and personalized texture maps.
- We propose an efficient fine-tuning strategy for training an ID-driven extension architecture for StableDiffusion, utilizing only a small-scale training dataset.
- We utilize our method to produce a new dataset, containing around 5k UVMap-ID image pairs, and will make it publicly available. Our small-scale attribute-balanced training dataset, the larger-scale dataset, and metrics for textures play a bridging role in guiding subsequent work in this field.

## 2 RELATED WORK

**UV-Map Generative Model.** This model aims to generate diverse textures based on the generative models, such as Generative Adversarial Networks [10], Diffusion Models [13, 43]. Existing works utilize this technique in the 3D face reconstruction with the 3D morphable model (3DMM) [3] or human reconstruction with the SMPL [29]. For face texture generation, GANFIT [9] first uses 10,000 high-resolution textures to train the GAN generator, then takes this GAN generator as the statistical parametric representation of the facial texture in the fitting progress. To avoid the training using the limited numbers and diversity of texture map, StyleUV [25] integrates the 2D image fitting and rendering stages into the adversarial networks. Additionally, some methods focus on contributing the 3D facial UV-texture datasets, such as Facescape [55], and FFHQ-UV [1]. For human texture generation, most of the works learn to recover the full texture from a single human image. The Re-Identification metric as supervised in this task is proposed [45]. To further improve the quality of texture generation, Zhao. et al [59] introduce

a consistency learning to enforce the cross-view consistency of texture prediction during training. Texformer [52] introduces the transformer architecture to exploit global information of the input, effectively facilitating higher-quality texture generation. Different from these methods without using any ground-truth 3D textures, Verica. et al [24] non-rigidly registers the SMPL model to thousands of 3D scans, and encoders the appearances as texture maps. And theses 3D textures are used to train a texture completed model. However, these mentioned methods cannot support diverse and text-guided texture generation. The most related work to ours is SMPLitex [5]. Motivated by the Dreambooth [37], SMPLitex utilizes a few texture maps to fine-tune the pretrained text-guided diffusion model to enable the textures inpainting and text-guided texture generation task. Compared to SMPLitex, our method supports both text-guided and ID-driven personalized texture generation.

**Text-to-3D Avatar Generation.** Text-guided 3D content generation has achieved great success with the development of 3D representation methods and generative models. Lots of methods utilize the frozen image-text joint embedding models from CLIP [33] to optimize the underlined 3D representation, such as NeRF [30] where some of them work on generation for general 3D object [18, 31, 40, 50, 54], or human Avatar [14, 16]. The most famous work is Dream Fields [18] which first demonstrated the effectiveness of combining the CLIP model and NeRF representation for 3D object creation, but 3D objects produced by this approach tend to lack realism and accuracy. DreamFusion [32] introduces Score Distillation Sampling (SDS) loss which is based on probability density distillation that enables the use of a pretrained 2D diffusion model as a prior for optimization of a parametric NeRF representation. By using SDS loss instead of CLIP, DreamFusion generates high-quality coherent 3D objects while aligning with the given text prompt. Recently, many similar methods with SDS loss have occurred to improve text-to-3D results in various aspects, such as enhancing the realism of rendering with detailed geometry [6], solving the multiple-view inconsistency problem [27, 42] or using variational score distillation (VSD) [47] method instead of SDS to improve the fidelity and diversity of 3D content generation. However, high-quality human avatars remain a challenge due to the complexity of the human body's shape, pose, and appearance. To make the avatar animatitable, DreamAvatar [4] and AvatarCraft [19] integrate the SMPL prior into the NeRF or SDF representation with a deformable field. To improve the avatar's quality and avoid the cartoon-like appearance, DreamHuman [23] uses a spherical harmonics lighting model instead of diffuse reflectance model and additionally optimizes a spherical harmonics coefficients; HumanNorm [17] introduces a normal diffusion model to enhances the diffusion model's understanding of 3D geometry to further improve the texture and geometry's quality. More recently, HumanGaussian [28] integrates 3D Gaussian representation instead of NeRF into 3D Human Avatar generation to reduce training time. Compared with these text-to-3D works, we focus on achieving a controllable texture generation but don't care about the generation of geometry.

**Text-Driven Personalized Diffusion Models.** Diffusion model [13, 43], is a class of generative modeling in which it iteratively transforms noises to samples simulating the true data distribution. Diffusion models generally outperformed other traditional methods, such as GANs, due to the fact that the output quality has been notably

improved across diverse domains. Diffusion models are widely used for text-to-image generation [34, 36, 38], and also stand out supporting more cross-model tasks [2, 35, 53]. One of the foundation works, Stable diffusion [36], applies the diffusion process on latent space, reducing training computation while preserving quality. While other methods, such as Imagen [38] and DALL-E2 [34], generate samples directed over pixel space, have also proven effective. Finetune-wise, DreamBooth [37] and LoRA [15] introduces a subject-driven training approach, enabling text controls, and offers a compelling feature for precise personalizing. Text Inversion [8] and VideoBooth [20] suggest an alternative solution via latent inversion before editing. Another class of methods [7, 46, 48, 49, 51, 56–58, 60] extends the model with additional networks to extract and adopt conditional inputs that guide the generation. Representatively, IP-Adapter [56] introduces a decoupled U-Net that injects conditional hidden features to the original diffusion U-Net, achieving an accurate control from the reference input. Some concurrent 2D methods such as Instant-ID [46], Infinite-ID [49] and SSR-Encoder [58], also attracted lots of attention. In this work, we share goals similar to IP-Adapter and Instant-ID, focusing on 3D human texture rather than 2D generation.

## 3 METHODS

Given a reference portrait describing the facial appearance (Face ID) of the target individual, our model aims to generate a texture that aligns with the facial appearance of the target person and fits the structure of the UV map defined by SMPL. In this section, we first provide a brief introduction to Denoising Diffusion Probabilistic Models [13] in Section 3.1, laying the foundational framework and network architecture for our method. Subsequently, detailed explanations of design specifics are presented in Section 3.2. Then, we will explain the pipeline we use to build the dataset in Section 3.3. Finally, we introduce some metrics for UV textures in Section 3.4.

## 3.1 Preliminary: Denoising Diffusion Probabilistic Models

The denoising diffusion probabilistic models operate by simulating a forward process that adds noise to an image or its latent representation over a series of time steps, transforming them into Gaussian noise. Conversely, the reverse process seeks to recover the original image or latent representation by iterative denoising. This bidirectional process is key to the diffusion models' ability to generate high-fidelity images. Our work leverages Stable Diffusion (SD), a pertrained generative model that could generate high-quality images from a text prompt. Specifically, given an image $x$, SD first uses a pretrained autoencoder to encode $x$ into latent: $z = \mathcal{E}(x)$. Then, noise is gradually added to $z$ over a sequence of $T$ steps, transitioning the data distribution from the original data distribution to a Gaussian Noise distribution, and the noise added forward a Markov chain of conditional Gaussian distributions defines the process:

$$q(z_t|z_{t-1}) = \mathcal{N}(z_t; \sqrt{1 - \beta_t}z_{t-1}, \beta_t I),$$

where $\beta_t$ is the variance schedule. During training, the denoising u-net $\epsilon_\theta$ of SD aims to learn to reconstruct the original latent $z$ from the noise, modeled by:

$$p_\theta(z_{t-1}|z_t) = \mathcal{N}(z_{t-1}; \mu_\theta(z_t, t), \sigma_\theta^2(z_t, t)\mathbf{I}),$$

and the learning objective is defined as follows:

$$L(\theta) = \mathbb{E}_{z_t, c, \epsilon, t}\left[||\epsilon - \epsilon_\theta(z_t, c, t)||^2\right],$$

where $c$ represents text conditional embeddings.

## 3.2 Fine-Tuning Text-to-Image Models for ID-Driven UV Map Generation

Fig. 2 provides the pipeline of our proposed approach. The initial input to the pipeline consists of random noise and a reference portrait. Our text-to-image model is configured based on the design of SD, employing the same framework and trained weights of SD. Motivated by DreamBooth [37], we propose to utilize the finetuning strategy with a prior preservation loss (Fig. 2 (Left)) applying to text-to-image diffusion architecture integrating with a face fusion module (Fig. 2 (Right)).

*3.2.1 Face Fusion Module.* To enable Stable Diffusion to accept additional image information, (i.e., the portraits), the previous methods mainly leverages the CLIP image encoder, either directly substituting the CLIP text encoder or through decoupled cross-attention mechanism to separate cross-attention layers for text features and image features [34, 56]. Nevertheless, the CLIP image encoder is constrained by its operation on images of lower resolution, which particularly impacts its efficacy in encoding face images by failing to encapsulate comprehensive details. Moreover, CLIP's architecture, fundamentally designed to align semantic features between text and images, mainly focuses on high-level feature correspondence. This orientation towards semantic feature matching inadvertently results in a dilution of finer, detailed features during the encoding process, posing a challenge for applications requiring precise detail retention. Hence, we propose to use the face embedding extracted by the face recognition models and linear projection layers to provide SD with human face information. Also, to preserve the original model's ability to process text information while integrating image information, we adopt the decoupled cross-attention mechanism [56], ensuring a seamless blend of both modalities. Given query feature $Z$, image feature $c_i$ and the text feature $c_t$, the output $Z'$ of decoupled cross-attention layers is:

$$Z' = \text{softmax}(\frac{QK^T}{\sqrt{d_k}})V + \text{softmax}(\frac{Q(K')^T}{\sqrt{d_k}})V',$$

where $Q = ZW_q$, $K = c_t W_k$, $V = c_t W_v$, $K' = c_t W_k'$, $V' = c_t W_v'$, and the $W_q$, $W_k$, $W_v$, $W_k'$ and $W_v'$ are learnable parameters of the projection layers. Similar fusion modules have been utilized in some concurrent 2D methods [46, 49].

*3.2.2 Prior Preservation Loss.* We observed that when using "UV texture map" as the text prompt, SD often fails to generate any correct UV maps. This is likely because SD is trained on data scraped from the internet, where real UV texture maps are rarely found in the training resources. Also, our goal is to generate images with a small training set (about 750 images in our dataset), each featuring different facial characteristics of individuals, and generating accurate faces has always been a weakness of SD. Additionally, our input incorporates extra face image information, and during fine-tuning, we would like to ensure our model does not lose SD's original capability to correctly process textual information. To this

**Figure 2: The left side of the figure shows the overview of our proposed pipeline. Given a reference image as face ID, we utilize a pre-trained text-to-image diffusion model, where the input is a combination of a noised UV Map and text prompt of a unique identifier and characteristics of the portrait where "A [S] Texturemap of [P]," where [S] is a unique identifier and [P] represents the race and gender. To maintain the quality of images generated by the pre-trained model and effectively process textual features, we adopt a prior preservation loss. The right side of the figure shows the detailed architecture of our model, where facial information is mapped to the same dimensions as text embeddings through a facial recognition model and face projection layers. Subsequently, we merge facial and textual information via decoupled cross-attention, which is then integrated into the pre-trained text-to-image model.**

end, we introduced prior preservation loss, as proposed in Dreambooth [37], to ensure the model retains its generalization ability and does not overfit the few-shot examples provided during the personalization process.

However, our objectives differ fundamentally from Dreambooth in two ways. Firstly, Dreambooth targets subject-driven generation, whereas our model aims at generating specific formats of images, the UV texture maps. This leads to a situation where Dreambooth requires re-fine-tuning the entire SD for each subject, while our model, after training, can generate corresponding UV maps for any input face ID. This distinction arises because, in DreamBooth, one unique identifier represents a single unique subject, whereas our unique identifier [S] denotes one unique kind of image structure (UV Map defined by SMPL). Secondly, we added extra facial information [P] to our text prompts during training to further preserve the original capabilities of the text encoder, enabling it to effectively parse attributes such as race and gender. For detailed experiments, please refer to Section 4.4

Formally, the training loss of our model is defined as:

$$L(\theta) = \mathbb{E}_{z_t, c, \epsilon, t} \left[ ||\epsilon - \epsilon_\theta(z_t, c, t)||^2 \right]$$
$$+ \mathbb{E}_{z_t, c', \epsilon, t} \left[ ||\epsilon_{\mathrm{pr}} - \epsilon_\theta(z_t, c', t)||^2 \right],$$

where $c'$ is a fixed conditional text prompt "a texturemap" and $\epsilon_{\mathrm{pr}}$ is the generate data using the frozen diffusion model with $c'$.

## 3.3 Dataset

**Training Dataset** In this part, we describe the process of constructing our dataset, which is centered around the generation of high-quality and diverse UV texture maps for digital human models. Our approach can be segmented into three stages:

1) Celebrity Selection: In the initial phase of our dataset creation, we aimed for a balanced and inclusive representation by employing OpenAI's ChatGPT to generate a list of 150 celebrities. Our selection was structured to include equal representation across three ethnic groups: African American, Asian, and White, with 50 celebrities from each group. To further enhance the diversity and applicability of our dataset, we ensured gender balance within each ethnic category, selecting 25 male and 25 female celebrities. We use celebrities because SMPLitex accepts only text input, and celebrity portraits are readily available. This approach allows us to link names, portraits, and corresponding UV texture maps effectively.

2) UV Texture Map Generation: We employed SMPLitex to generate UV texture maps for each of the selected celebrities. This process resulted in 50 UV texture maps per celebrity, totaling 7,500 initial texture maps.

3) Manual Selection: To ensure the highest quality and relevance for our dataset, we manually reviewed the generated UV texture maps and selected 5 maps per celebrity that best met our predefined criteria. These criteria included clarity, detail accuracy, and representation quality of ethnic features. This manual selection process narrowed our dataset to 750 UV texture maps with 5 UV texture maps per ID.

**A New Dataset: CelebA-HQ-UV** We utilize our method with personalized generation to produce a new dataset, which contains 5k UVMap-ID pairs. Specifically, we select 5000 high-resolution face images from CelebA-HQ [21] as reference image IDs of our methods. For every ID, our method produces 10 textures and selects 2 by the evaluation of multiple aspects, i.e., the quality of textures, the preservation of UV structure, and the preservation of face ID. Fig. 3 shows some results using three face IDs from CelebA-HQ. We refer to this dataset as CelebA-HQ-UV, and will make it publicly available.

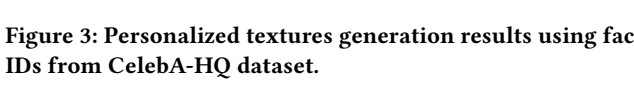

Figure 3: Personalized textures generation results using face IDs from CelebA-HQ dataset.

Note that we define a list of text prompts for these generations which will be introduced in the supplementary material.

### 3.4 Metrics

As previously mentioned, assessing the quality of generated textures within the UV space defined by SMPL poses a significant challenge, especially within the scope of our personalized generation task. In this paper, we introduced four metrics to evaluate the quality of the generated textures from multiple aspects: Inception Scores [39] to evaluate the fidelity and diversity, Semantic Structure Preservation (SSP) to evaluate structure preservation of UV space defined by SMPL [29], Deep Face Recognition (DFR) to evaluate Face ID preservation and CLIP-Text (CLIPT) [20, 48] score to evaluate the text-image alignment.

**Inception Score (IS) on UV textures and rendered results** The Inception Score (IS) and Fréchet Inception distance [12] are widely utilized metrics for evaluating the diversity and quality of 2D images generated by generative models. FID is a well-established measure that compares the inception similarity score between distributions of generated and real images. One key distinction between IS and FID is that IS is computed solely using fake samples, eliminating the need for real samples in its calculation. Due to the lack of real sample distribution, we employ the IS to directly evaluate the quality of 5000 generated textures rather than FID. We refer to IS on textures of UV space as IS (UV). Additionally, we render these textures into 2D space by applying them to the SMPL Mesh. Subsequently, we utilize IS to evaluate the quality of 5000 rendered human images in 2D space. We refer to this type of IS as IS (R).

**Semantic Structure Preservation (SSP)** To assess the preservation of UV structures in generated textures, we introduce a novel metric termed Semantic Structure Preservation (SSP). Notably, we have observed instances where the generated textures from SMPLitex [5] may not faithfully retain these underlying structures, as illustrated in Fig. 4. The SSP metric is designed to quantify this preservation. We leverage off-the-shelf human parsing techniques [26]

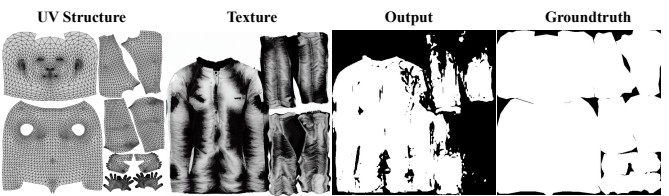

Figure 4: It shows UV structures, textures from SMPLitex, extracted semantic segmentation, and semantic groundtruth from left to right.

to extract semantic segmentation from the generated images and then compare it with ground truth segmentation (Fig. 4 (right)). We conduct this comparison across a dataset comprising 1000 images and compute the mean difference as the SSP score.

**Deep Face Recognition (DFR)** To assess the preservation of identity (ID) within textures, a crucial aspect of personalized image generation tasks, we propose employing Deep Face Recognition (DFR) methods to quantify the similarity between generated textures and reference facial images. Specifically, we leverage the off-the-shelf tool [41] to do face recognition between the textures and image ID. We use 10 face IDs, and 100 samples for every ID and report the successful numbers. We refer to this metric as the DFR score which is reported as a measure of the preservation of identity within the generated textures.

**CLIP-Text (CLIPT)** To measure the alignment of the generated textures and given text prompts, we use the CLIP-Text (CLIPT) score followed by 2D methods [20, 48]. This metric is calculated using the cosine similarity of the CLIP text embeddings of the given text prompts and CLIP image embeddings of the generated textures. We compute the CLIPT score using 1000 text-prompt pairs.

## 4 EXPERIMENTS

### 4.1 Training Details

Our experiments are based on the Realistic_Vision_V4 model, which is further fine-tuned on Stable Diffusion v_1.5 [36], and could produce more photorealistic images. Additionally, we utilize the buffalo_l pre-trained face recognition model from SCRFD [11], and pre-trained projection layers from [56]. The experimental code is developed using the HuggingFace Diffusers library [44]. During training, we fine-tune the entire U-Net, text encoder and face projection layers, and keep the VAE encoder and decoder of Stable Diffusion frozen. The UVMap-ID training is conducted on a single machine equipped with an A40 GPU for 1500 steps, with a batch size of 2. We employ the AdamW optimizer [22] with a fixed learning rate of 1e-6 and a weight decay of 0.01. Our dataset comprises images with a resolution of 512 × 512, hence we generate images at this resolution during training. In the inference phase, we use a 50-step DDIM sampler [43] and set the classifier-free guidance scale to 7.5.

### 4.2 Baselines

We take the texture generation model SMPLitex [5] as the baseline. And all results from SMPLitex are produced from their released code

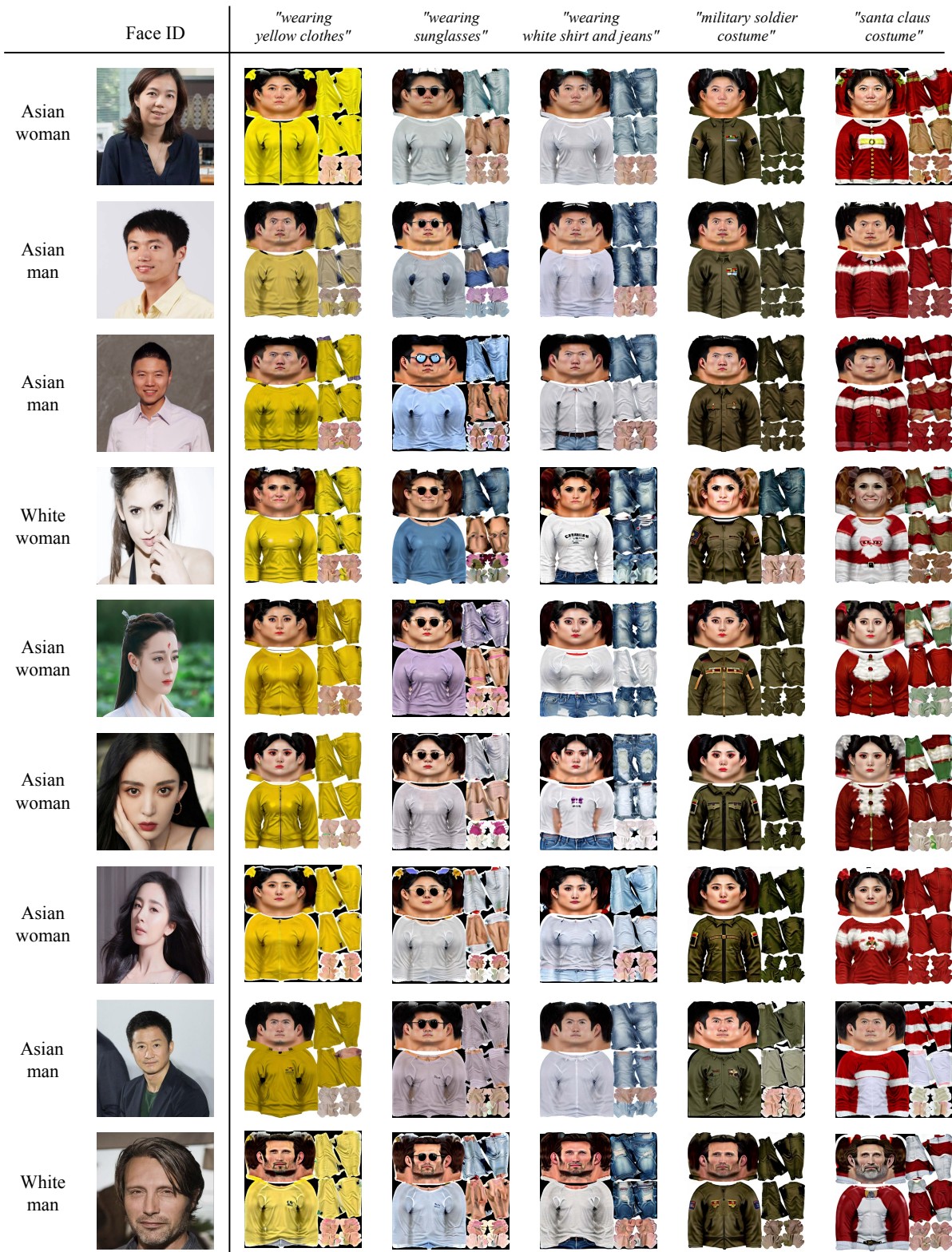

**Figure 5: Our personalized generation results. The 1st column shows reference faces, obtained from the website, and not existing in our training set.**

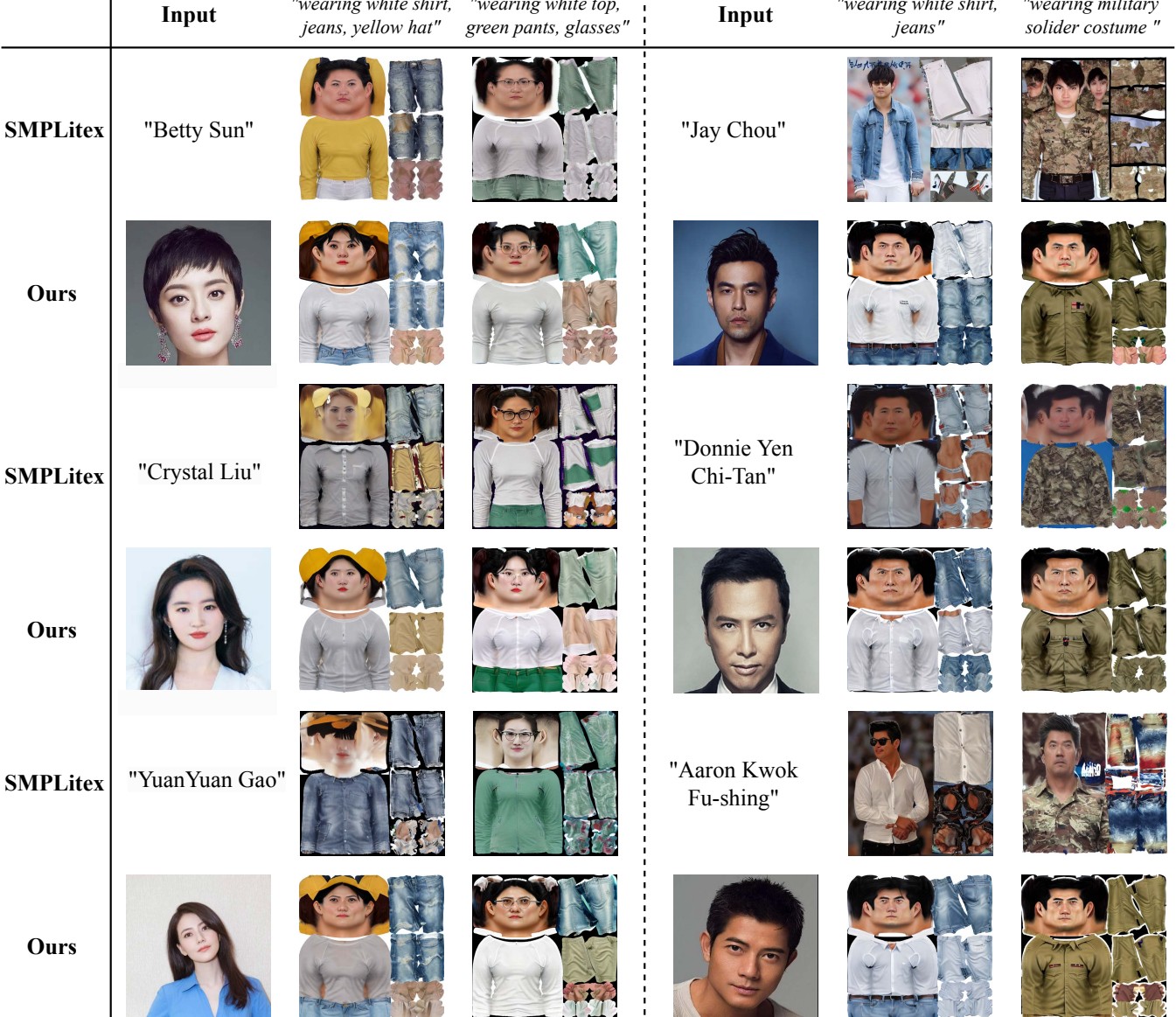

**Figure 6: Comparsion with SMPLitex [5] results. SMPLitex is not an image ID-driven method. Thus, we provided these celebrities' names in the test prompts for SMPLitex, but not for ours. Taking "Betty Sun" as an example (upper-left corner), the test prompt of SMPLitex is "a texturemap of Betty Sun wearing...", and our test prompt is "a texturemap of Asian woman wearing...". Note that image IDs are not existing in our training data.**

and pretrained model. SMPLitex does not support image-driven personalized generation. Thus, we provide image ID's name in the text prompts for SMPLitex, but not for our method.

### 4.3 Comparisons

Fig. 5 shows diverse personalized texture generation results from our methods. Our reference face IDs (1st column images) are collected from a diverse range of sources on the website, thus encompassing a wide variety of characteristics, including different ethnicities, genders, occupations, levels of fame, and even facial

| Methods | IS (R) ↑ | IS (UV) ↑ | SSP ↓ | CLIPT ↑ | DFR ↑ |
|---|---|---|---|---|---|
| SMPLitex [5] | 1.46 ± 0.020 | **1.95 ± 0.049** | 10.45 | **29.40** | 62 |
| UVMap-ID | **1.78 ± 0.020** | 1.89 ± 0.027 | **8.46** | 29.12 | **792** |

**Table 1: Quantitative results using four metrics: inception scores on rendered images (IS (R)), inception scores on UV maps (IS (UV)), Semantic Structure Preservation (SSP), CLIP Text (CLIPT), Deep Face Recognition (DFR).**

| Methods | DFR ↑ |
|---|---|
| UVMap-ID $w/o$ "Race and Gender" | 436 |
| UVMap-ID $w/$ "Race and Gender" | **792** |

**Table 2: Ablation Study for "Race and Gender" label.**

| Methods | IS (R) ↑ | IS (UV) ↑ | SSP ↓ | CLIPT ↑ | DFR ↑ |
|---|---|---|---|---|---|
| UVMap-ID (1) | **1.88 ± 0.028** | **2.03 ± 0.039** | 10.59 | 29.09 | 734 |
| UVMap-ID (2) | 1.78 ± 0.020 | 1.89 ± 0.027 | **8.46** | 29.12 | 792 |
| UVMap-ID (5) | 1.55 ± 0.017 | 1.55 ± 0.084 | 8.74 | **29.27** | 798 |

**Table 3: Ablation studies of Training data. UVMap-ID ($N$) denotes the number ($N$) of textures for each ID in the training stage.**

poses. As shown in the 2nd-6th columns of Fig. 5, our generated UV textures effectively preserve the identity features of these reference face IDs, demonstrating the effectiveness and robustness of our methods in personalized generation. Moreover, our method also achieves accurate text-driven controllable generation.

We conducted visualization comparisons with SMPLitex [5], as depicted in Fig. 6. Notably, SMPLitex is not an image-driven method. Therefore, while we utilized some well-known celebrities as image IDs and provided their names in text prompts for SMPLitex, we deliberately omitted this information for our method to ensure a fairer comparison. Remarkably, our results exhibit a higher degree of similarity in face ID preservation compared to SMPLitex, underscoring the superiority of our method in maintaining identity features during personalized texture generation. Moreover, our approach also demonstrates superior structural preservation compared to SMPLitex, as evidenced by the "Jay Chou" row (Top-Right).

Quantitative results using four metrics are shown in Table 1. We observe that SMPLitex achieves better IS (UV) scores than our method. We attribute this to the fact that our approach is image-driven, which means that the provided reference ID constrains the diversity of generated images, a crucial aspect of IS. In contrast, our method achieves a higher IS (R) than SMPLitex. As mentioned, SMPLitex often struggles to preserve UV structures effectively, resulting in unrealistic renderings. The comparison of structure preservation can be validated by our achieved superior SSP score. Moreover, our DFR score significantly outperforms the Baseline, validating that our method achieves better similarity to the target ID in personalized texture generation tasks. Additionally, the high success rate of 837 out of 1000 demonstrates the robustness of our method to reference images. Furthermore, we observe that our CLIPT score is comparable to the baseline, indicating that the "image prompt" generated by our image encoder does not significantly affect the control capability of the text prompt.

### 4.4 Ablation Studies

**"Race and Gender" in prompts** As shown in Fig. 7, we analyze the impact of including race and gender labels in prompts during training, assessing how this additional information affects generative model performance. As indicated in Table 2, incorporating race and gender labels significantly enhances the model's DFR score compared to the version without these labels (UVMap-ID $w/o$ "Race

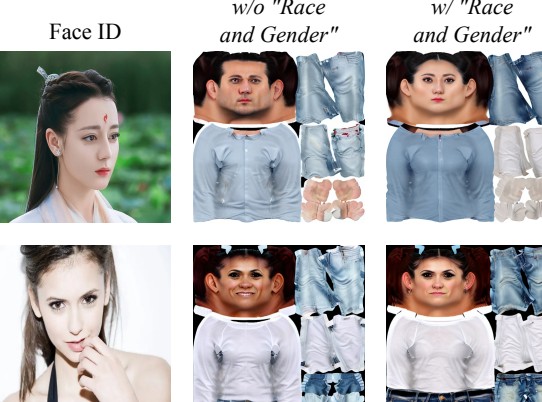

| Face ID | w/o "Race and Gender" | w/ "Race and Gender" |
|---|---|---|

**Figure 7: Qualitative ablation studies of between $w/o$ and $w/$ "Race and Gender" labels. The 1st-row results show our full method preserves the "Gender" attribute and the 2nd-row results show our full method preserves the "Race" attribute.**

and Gender"). This indicates that the facial recognition model we use focuses more on the structural information of the human face, while the label supplements the missing information such as skin color.

**Training Data** In this part, we explore the impact of varying the number of UV maps used per image ID during training. Our model, UVMap-ID, is evaluated using a consistent training strategy, except that each image ID in the training dataset is processed using 1, 2, or 5 UV maps. These setups are denoted as UVMap-ID (1), UVMap-ID (2), and UVMap-ID (5) respectively.

Table 3 highlights the performance metrics across these configurations. Based on the results shown in Table 3, we have chosen UVMap-ID (2) as our base model. This configuration utilizes two UV maps, which provide a diverse dataset sufficient to capture the critical variations in facial features, without overloading the pre-trained model. UVMap-ID (2) strikes a balance, delivering remarkable realism in image generation while effectively maintaining the identity of reference images.

## 5 CONCLUSIONS

In this paper, we introduce UVMap-ID, the first method for ID-driven personalized texture generation. UVMap-ID takes the StableDiffusion as the backbone and extends it with an additional face fusion module. Moreover, our method is a highly efficient model with only several hours fine-tuning strategy on a small-scale dataset. Additionally, we also explore the evaluation of quality for UV textures and introduce some corresponding metrics. Finally, with user provided face images, our method can automatically create high-quality UV textures with the preservation of face ID while enabling text-driven controls, which is a very available application for 3D avatar creation in compute graphics fields. By using our method, we create a new dataset, CelebA-HQ-UV, comprising textures and face ID pairs. This dataset will be shared with the community to facilitate further research. We desire to explore the interactive editing of textures in the future.

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
