# OpenReview forum: "UVMap-ID: A Controllable and Personalized UV Map Generative Model"
_acmmm.org/ACMMM/2024/Conference — MM2024 Poster_

### Official Review · Reviewer_Ez8W · 2024-05-24

**Rating:** 4
**Confidence:** 2

**Summary:**

This paper is based on these two questions: how to generate personalized texture maps for any given face image, and how to define and evaluate the quality of these generated texture maps. This paper proposes a new method for UV map generation that is controlled and personalized. A balanced fine-tuned dataset is also proposed. A variety of evaluation metrics are introduced.

**Strengths:**

1.	The work in this paper is groundbreaking.
2.	The generative performance demonstrated in this paper is excellent.

**Limitations:**

1.	The generation about controlled and personalized does not seem to be well explained in the paper.
2.	The motivation for the Cross-Attention mechanism is not sufficiently clearly described.
3.	Figure.2 is not well drawn and the innovation "Controllable and Personalized" is not clearly visible.
4.	The comparative experiments in this paper are inadequate, with only one method, SMPLitex.

**Suitability:**

3

---

### Official Review · Reviewer_ustD · 2024-05-25

**Rating:** 1
**Confidence:** 2

**Summary:**

This paper proposes that the UVMap-ID model can generate personalized texture maps based on a given facial image (Face ID). This method is different from traditional methods that require large-scale dataset training. UVMap-ID effectively achieves personalized texture generation by fine-tuning small-scale, attribute-balanced datasets. The main contributions include the introduction of new evaluation metrics in the paper, such as Inception Score, Semantic Structure Preservation (SSP), Deep Face Recognition (DFR), and CLIP Text (CLIPT), providing a multidimensional quantitative method for the quality evaluation of UV textures.

**Strengths:**

The main contributions include the introduction of new evaluation metrics in the paper, such as Inception Score, Semantic Structure Preservation (SSP), Deep Face Recognition (DFR), and CLIP Text (CLIPT), providing a multidimensional quantitative method for the quality evaluation of UV textures.

**Limitations:**

1. It is recommended that the author provide a more detailed introduction to the construction process of the dataset in the methods section, including how to select and generate texture maps, as well as how to ensure the diversity and balance of the dataset.
2. The paper mentioned the training performance of the model on small-scale datasets, but the generalization ability of the model on unseen facial images has not been fully verified. Suggest conducting additional experiments to evaluate the model's generalization ability.
3. It is recommended that the author consider conducting user research to evaluate the user experience and acceptance of the generated UV texture in practical applications.
4. The paper did not mention the computational efficiency of model training and texture generation. Given the real-time requirements of 3D graphics applications, it is recommended that the author provide more information on model efficiency.
5. Although the paper proposes multiple advantages of UVMap-ID, it is also important for any research to discuss the limitations of the model and potential improvement directions. Suggest the author to add an analysis of the current limitations of the model in the discussion section.

**Suitability:**

2

---

### Official Review · Reviewer_9BFs · 2024-05-29

**Rating:** 3
**Confidence:** 3

**Summary:**

The author divides the process of generating a 3D avatar from text and human face into two parts: 1. Converting text descriptions and face into UV maps, and 2. Transforming UV maps into a 3D model. This paper primarily addresses the first part, presenting a solution that employs diffusion models along with specifically designed text prompts.

**Strengths:**

The writing is very clear, making the complex process easy to understand.

It's encouraging to see research focusing on generative 3D models, which is a promising area in the field.

**Limitations:**

The proposed pipeline is not very convincing to me. For instance, the proposed method outputs a fixed size of UV maps. How can this approach accommodate different body shapes, such as varying heights of people? This requires further discussion. Additionally, a comparison of the proposed pipeline with single-view 3D shape generation should be included to highlight its advantages.

The author claims an even representation of celebrities from African American, Asian, and White backgrounds. However, in Figure 5, Asian faces are in the majority. Is there a reason for this discrepancy? Could it be due to dataset bias?

Furthermore, the author only compares the proposed method with one baseline. More discussion on this point is necessary to strengthen the validation of the proposed approach.

**Suitability:**

2

---

### Official Review · Reviewer_quUQ · 2024-05-31

**Rating:** 2
**Confidence:** 2

**Summary:**

This paper proposes a controllable and personalized UV map generative model, which is based on pre-trained text-to-image diffusion model,  to synthesize personalized texture maps and achieve IDdriven customized generation.

**Strengths:**

1. The concept of synthesizing diverse and personalized texture maps is relatively novel.
2. This paper produces a new dataset, which contains 5k UVMap-ID pairs.

**Limitations:**

1. This paper's contribution to methodology is minimal. Although the proposed method has many capabilities, it is the result of putting together many known network architecture for feature extraction or fusion  and loss terms in the objective that achieves each capability. In my opinion, the good experimental results are due to the powerful generative capabilities of the diffusion model, and the loss function is also borrowed from others' work.
2. Did you create the small-scale attribute-balanced dataset?
3. One of the main contributions of this paper is the creation of a dataset. However, without relevant experiments on this dataset to lay the groundwork for future work, what is the significance of generating the dataset?

**Suitability:**

2

---

### Meta-Review · Area_Chair_eLK7 · 2024-07-02

**Recommendation:** Accept (Poster)
**Confidence:** 4

**Metareview:**

Pros:
- This paper propose to turn the T2I model into a controllable and personalized UV map generation model, which is interesting.
- The T2I model is only required to be fine-tuned on a small scale dataset to achieve the above goal.
- The experiments show the promising results.
- A new dataset and evaluation metrics are proposed.

Cons:
- The motivation of the key idea needs further explained.
- The writing needs further refinement for more easy understanding of some details, such as on the datasets.
- Figure 2 needs improvement.